# Berberine Derivatives as *Pseudomonas aeruginosa* MexXY-OprM Inhibitors: Activity and In Silico Insights

**DOI:** 10.3390/molecules26216644

**Published:** 2021-11-02

**Authors:** Giorgia Giorgini, Gianmarco Mangiaterra, Nicholas Cedraro, Emiliano Laudadio, Giulia Sabbatini, Mattia Cantarini, Cristina Minnelli, Giovanna Mobbili, Emanuela Frangipani, Francesca Biavasco, Roberta Galeazzi

**Affiliations:** 1Department of Life and Environmental Sciences, Polytechnic University of Marche, Via Brecce Bianche, 60131 Ancona, Italy; giorgia.giorgini@pm.univpm.it (G.G.); g.mangiaterra@staff.univpm.it (G.M.); n.cedraro@staff.univpm.it (N.C.); giulia.sabbatini@univpm.it (G.S.); m.cantarini@pm.univpm.it (M.C.); c.minnelli@staff.univpm.it (C.M.); g.mobbili@staff.univpm.it (G.M.); f.biavasco@univpm.it (F.B.); 2Department of Materials, Environmental Sciences and Urban Planning, Polytechnic University of Marche, Via Brecce Bianche, 60131 Ancona, Italy; e.laudadio@staff.univpm.it; 3Department of Biomolecular Sciences, University of Urbino Carlo Bo, 61029 Urbino, Italy; emanuela.frangipani@uniurb.it

**Keywords:** efflux pump inhibitors, *Pseudomonas aeruginosa*, berberine derivatives, molecular modeling, multidrug resistance

## Abstract

The natural alkaloid berberine has been demonstrated to inhibit the *Pseudomonas aeruginosa* multidrug efflux system MexXY-OprM, which is responsible for tobramycin extrusion by binding the inner membrane transporter MexY. To find a structure with improved inhibitory activity, we compared by molecular dynamics investigations the binding affinity of berberine and three aromatic substituents towards the three polymorphic sequences of MexY found in *P. aeruginosa* (PAO1, PA7, and PA14). The synergy of the combinations of berberine or berberine derivatives/tobramycin against the same strains was then evaluated by checkerboard and time-kill assays. The in silico analysis evidenced different binding modes depending on both the structure of the berberine derivative and the specific MexY polymorphism. In vitro assays showed an evident MIC reduction (32-fold and 16-fold, respectively) and a 2–3 log greater killing effect after 2 h of exposure to the combinations of 13-(2-methylbenzyl)- and 13-(4-methylbenzyl)-berberine with tobramycin against the tobramycin-resistant strain PA7, a milder synergy (a 4-fold MIC reduction) against PAO1 and PA14, and no synergy against the Δ*mexXY* strain K1525, confirming the MexY-specific binding and the computational results. These berberine derivatives could thus be considered new hit compounds to select more effective berberine substitutions and their common path of interaction with MexY as the starting point for the rational design of novel MexXY-OprM inhibitors.

## 1. Introduction

*Pseudomonas aeruginosa* is an opportunistic pathogen and a frequent cause of life-threatening infections in high-risk patients [1,2,3], including those with a compromised immune system due to underlying diseases such as cancer or cystic fibrosis [4]. *P. aeruginosa* infections are difficult to eradicate due to the large number of already-known intrinsic and acquired antibiotic resistance mechanisms [5]. In particular, the chromosomally encoded efflux pumps are responsible for the multidrug-resistant (MDR) phenotype and their overexpression largely contributes to drug tolerance and failure of antibiotic treatment [5,6]. MexXY-OprM, belonging to the resistance–nodulation–cell division (RND) family, is considered the main resistance mechanism towards aminoglycosides in *P. aeruginosa*; mutations in its regulatory gene *mexZ* [6] can lead to overexpression and hamper antibiotic treatments. This efflux pump also represents the best example of adaptive resistance, showing a transient hyperexpression in the presence of antibiotics and a basal expression when the drug is removed [7,8].

The MexXY-OprM system extrudes hydrophilic compounds, including aminoglycosides such as tobramycin. This substrate specificity is not shared by the better-known MexAB-OprM efflux pump due to the different aminoacidic composition of the substrate binding pockets, which are located in the inner membrane transporters MexY and MexB, respectively [9]. In previous studies, we optimized a full 3D model of the MexY structure using the highly similar models of the membrane channels *P. aeruginosa* MexB and *Escherichia coli* AcrB [9,10,11]. From a structural point of view, MexXY-OprM is a tripartite system characterized by a drug extrusion mechanism via a proton gradient as in all RND efflux systems [12]. MexY is the inner membrane protein, organized as a homotrimer presenting a transmembrane (TM) domain, with 12 alpha helices and a periplasmic domain, which is directly involved in the access, binding, and extrusion of substrates. The periplasmic domain further includes a porter domain (PD) and a TolC domain [13]. The PD is characterized by four other subdomains called PC1, PC2, PN1, and PN2, while the TolC domain is split into two subdomains named DC and DN (Figure 1). The substrate extrusion pathway is characterized by a functional rotation and a cycling conformational change in each protomer of the homotrimer [14] in three different states: loose (for the access stage), tight (for the binding stage), and open (for the extrusion stage). The PD of each protomer encloses two substrate binding pockets, the access or proximal (AP) and the binding or distant (DP) one. Both of them are involved in ligands’ translocation through a peristaltic motion as suggested for AcrB [15]. These two binding sites are separated by a G-loop (not shown in Figure 1), which regulates the passage between these two pockets [16]. Three different pathways can be described for the access of the substrates to the MexY transporter (channel CH) in analogy to AcrB EP [17]: (i) the entrance above TM7/TM8 helices from the outer leaflet of the inner membrane to the proximal and binding pocket (CH1); (ii) periplasmic, through the cleft via hydrophilic compounds between subdomains PC1 and PC2 (CH2); and (iii) through the vestibule between protomers into the central cavity (CH3) (Figure 1).

The MexY inner channel of different *P. aeruginosa* strains is characterized by extensive sequence polymorphisms, which can affect the binding modes of both substrates and efflux pump inhibitors (EPIs). These polymorphisms can explain the differences in the ability of MexY to extrude EPIs, and the study of the binding affinity in different bacterial strains can help to identify new molecules to be used as efficient EPIs.

Although the MexY inhibitory activity of the natural alkaloid berberine was extensively assessed by in silico modeling [11,12,18], the association between berberine and tobramycin resulted in a strain-dependent variable behavior [11]. Previous studies showed that the introduction of an aromatic substituent in position 13 can improve both the synergism of the combination berberine–fluconazole against *Candida albicans* [19] (Figure 1) and the antimicrobial activity of the alkaloid itself against multidrug-resistant (MDR) *Staphylococcus aureus* [20].

Starting from this evidence, we decided to assess the synergism of the three aromatic alkaloid derivatives with tobramycin [19,20]. Thus, we evaluated berberine and its derivatives’ potential activity in silico considering both affinity (free binding energy) and binding specificity through molecular docking and molecular dynamic simulations and carrying out in vitro microbiological assays. Moreover, we analyzed the aminoacidic sequence of MexY of the three reference *P. aeruginosa* strains PAO1, PA7, and PA14 to assess the involvement of specific polymorphisms in the synergistic effect of the combinations of the tested compounds with tobramycin.

## 2. Materials and Methods

### 2.1. Synthesis of Berberine Derivatives:

The berberine derivatives (Figure 1) were synthesized according to Kotani et al. [18] with minor changes. Briefly, after dropwise addition of the appropriate benzyl bromide (1.0 mmol) to a dihydroberberine (1 equiv) and KI (2 equiv) CH_3_CN (40 mL) solution, the reaction mixture was refluxed under stirring for 4 h. After filtration and solvent evaporation, the crude residue was purified by silica gel column chromatography using CHCl_3_/CH_3_OH (50:1) as the eluent. The characterization data of the compounds obtained were identical to those given in the literature [18,20].

### 2.2. Computational Methods

#### 2.2.1. Sequence Alignment and 3D Modeling for the Polymorphic MexY

The MexY aminoacidic sequence alignment was performed using the multiple sequence alignment method (MAFFT) [21] with default parameters. The aminoacidic sequence of each MexY protein was retrieved in FASTA format from the NCBI protein database. The 3D models of the three polymorphic MexY proteins belonging to *P. aeruginosa* PAO1 (NCBI code BAA34300.1), PA14 (NCBI code ABR84278.1), and PA7 (NCBI code QDL65075.1) were constructed following the comparative molecular modeling approach [2,11] using a X-ray crystallographic protein structure of the MexAB-OprM inner transporter as the template (pdb code: 2V50), since it shares the structural function with an identity percentage of 49.95% (PA7), 50.05% (PA14), and 47.17% (PAO1). The three models were built using the Swiss Model server [22] and, after generation, each new 3D protein structure was minimized in a homotrimeric system within the membrane of 500 1-palmitoyl-2-oleoyl-sn-glycero-3-phosphocholine lipids (POPC), thus reaching the nearest local minimum conformational energy on the potential energy surface (PES). This was performed using cycles of steepest descent and conjugate gradient algorithms implemented in the GROMACS v.2020.6 software [23,24] until the system converged. For each minimization step, the Force f and energy potential V of the molecular conformation were also computed. The models were validated using the protocol already reported in our previous work [11,25].

#### 2.2.2. Molecular Docking

The molecular docking simulation was performed using Autodock 4.2.1 (AD4) [26]. Each minimized protein was retrieved in its monomeric form, adding hydrogens to the structure, converting each pdb file into a pdbqt file, and coordinating it to include the calculated charges. The same approach was used for the ligands’ preparation, and, for each ligand–protein complex, the calculation of the electrostatic potential grid was performed with the Autogrid tool, setting a grid box with 50 Å^3^ focusing near the cleft site of the MexY protein, including PC1 and PC2 subdomains and two periplasmic loops of TM7 and TM8 helices. The genetic algorithm (GA) was used for the pose generations of each ligand, and the AMBER force-field-based scoring function was used for energy calculations as implemented in the docking software. The number of independent AD4 GA runs was increased up to 100, and the grid spacing was kept at 0.375.

#### 2.2.3. Molecular Dynamics

The MexY models together with berberine and its derivatives were oriented in the membrane through the OPM server [27]. The CHARMM GUI [28,29,30] was used to build a membrane bilayer system composed of 500 POPC molecules. MexY trimers with the docked ligands were generated properly surrounded by the lipid matrix. All the built models were appropriately solvated with water (about 10,000 molecules) and ions (to reach up to 0.15 M NaCl, adding 397 Na ions and 361 Cl ions to balance the trimer charge). CHARMM36 force field parameters [31] were used for all MD simulations together with the TIP3P [32] model for the solvent as implemented in GROMACS [23,24]. Berberine and its derivatives were parametrized within the CHARMM-GUI Ligand Reader and Modeler tools for the chosen force field [33] in their positive charged state (Figure 1). After the model minimization, six equilibration phases and MD simulations were carried out. The overall time of each MD simulation was settled to 50 ns, with a time-step of 0.002 ps. Periodic boundary conditions (PBCs) were applied in all directions using a neighbor searching grid type and setting at 1.4 nm the cut-off distance for the short-range neighbor list. Electrostatic interactions were taken into account by implementing a fast and smooth Particle-Mesh Ewald algorithm, with a 1.4 nm distance for the Coulomb cut-off [34].

### 2.3. Microbioloogical Methods

#### 2.3.1. Bacterial Strains, Plasmids, Media, and Chemicals

The *P. aeruginosa* laboratory strain K767 (PAO1), its Δ*mexXY* mutant K1525, and the plasmid pYM004, containing a copy of the *mexXY* operon, were kindly provided by Prof. Keith Poole (Queen’s University, Kingston, ON, Canada); the mutant strain was complemented by transferring the pYM004 plasmid by electroporation and selected on Luria–Bertani (LB) agar plates containing 200 µg/mL carbenicillin. *P. aeruginosa* PA14 was kindly provided by Prof. Olivier Jousson (Integrated Biology Center, University of Trento, Trento, Italy). The *P. aeruginosa* PA7 strain belongs to the Microbiology section strain collection of the Department of Biomolecular Sciences, University of Urbino “Carlo Bo”, Urbino, Italy, while *P. aeruginosa* ATCC 27853 belongs to the Microbiology section strain collection of the Department of Life and Environmental Sciences—Microbiology section, Polytechnic University of Marche, Ancona, Italy.

Carbenicillin and tobramycin were purchased from Sigma-Aldrich SRL (Milano, Italy), and all microbiological media were purchased from Oxoid SpA (Milano, Italy).

#### 2.3.2. Antimicrobial Combination Assays

##### Antimicrobial Susceptibility and Checkerboard Assays

*P. aeruginosa*’s antimicrobial susceptibility towards tobramycin and berberine derivatives was assessed by determining the Minimal Inhibitory Concentration (MIC) according to CLSI guidelines (2017) [35]. For the derivatives, a concentration of 320 µg/mL was selected as the highest, in order not to exceed 1.5% DMSO. The solvent itself was tested as well. Checkerboard assays were performed as previously described [11,36] using 2-fold dilutions of tobramycin (from 256 to 0.25 µg/mL when testing *P. aeruginosa* PA7 and from 16 to 0.01 µg/mL when testing *P. aeruginosa* PAO1 and PA14) and of berberine derivatives (from 320 to 10 µg/mL). A four-fold reduction of tobramycin’s MIC was considered indicative of synergism [11]. The reduction in tobramycin’s MIC in the presence of 80 µg/mL berberine was used as a reference value.

##### Time Killing Curves

Time killing assays were performed as previously described [11,36] using tobramycin concentrations corresponding to 1/2× the MIC, 1× the MIC, and 2× the MIC, alone or in combination with the different berberine derivatives at the concentration that was more synergic than berberine’s one in checkerboard assays. The survivor amount was determined by a plate count on LB agar after 0, 2, 4, 6, 8, and 24 h from drug(s) exposure. Each count was the average of the results of two technical duplicates.

## 3. Results

### 3.1. Computational Results

#### 3.1.1. MexY Polymorphisms in PAO1, PA7, and PA14

The MexY sequences of *P. aeruginosa* PAO1, PA7, and PA14 were aligned, and an identity matrix percentage is reported (Table 1).

To evaluate the aminoacidic variations between these sequences, a multisequence alignment with the MAFFT method was performed considering the PAO1–MexY complex as the reference sequence (Figure 2). Several aminoacidic substitutions were found in *P. aeruginosa* PA7 and PA14. Some of them (highlighted in red in Figure 2) were similar in both strains, while others were strain-specific (reported in green for *P. aeruginosa* PA7 and in blue for *P. aeruginosa* PA14).

Proline 862 of MexY–PAO1 is lacking in both MexY–PA7 and MexY–PA14, while residues 800, 858, 1037 (1036 in PA7, PA14), and 1040 (1039 in PA7, PA14) (marked in aquamarine) differed in all the three sequences. To better localize variant sites (Figure 2), the aminoacidic patterns that constitute the PC1–PC2–PN1–PN2 subdomains of the porter domain are indicated in this multisequence alignment (MSA). As mentioned above, this porter domain encloses both the access pocket (AP) and the deep pocket (DP), which are generally involved in the access and binding of the substrate, respectively, in its extrusion pathway. These subdomains in MexY variants were identified on the basis of MexB subdomains [37,38,39] through a multi-alignment sequence (Figure 2).

It is worth noting that the differences in aminoacidic composition, as represented in Figure 2, mainly affect the CH1 entrance cleft site that is adjacent to the AP.

#### 3.1.2. Molecular Docking of Berberine Derivatives

The different binding affinities of the three selected berberine derivatives towards the MexY proteins were firstly assessed, considering the aminoacidic variations in the MexY sequences in *P. aeruginosa* PAO1, PA7, and PA14, (Figure 3, Table 2). The docking of the potential EPIs was focused on the periplasmic site and the cleft site, which are involved in the extrusion of lipophilic, small-sized compounds and hydrophilic larger compounds, respectively [38].

Docking results were analyzed considering the best scoring poses for each ligand within MexY complexes in the three different considered strains and comparing the derivatives’ affinity with that of the parent compound berberine. All the tested compounds show a favorable binding energy in all the three complexes, with the best in silico results obtained for the MexY–PA7/ligand complexes for both berberine and its derivatives (Figure 3A).

From the docking focusing on the two periplasmic loops of TM7–TM8 helices, it emerges that in MexY–PA7 complexes all berberine derivatives adopt the same orientation inside the cavity (Figure 3A), with the aromatic moiety orientated toward the periplasmic side. On the contrary, inside the other two MexY complexes (PAO1, PA14), the ligands are positioned in different orientations and, comparing their binding affinities (Figure 3B, Table 2), none is more favorable than the corresponding one for the MexY–PA7 model.

Besides, in the MexY–PA14 complex, the p-CH_3_-berberine (green) and p-CF_3_-berberine (blue) are located with same orientation in the target site, while o-CH_3_-berberine (cyan) is located with an inverted opposite pose (Figure 3C). Analyzing in detail the berberine docking interactions (Table 2) in these three different complexes, we can point out the presence of the hydrophobic residues ILE^38^ PHE^560^ LEU^669^ LEU^561^ ALA^825^ e LEU^666^ that stabilize all MexY-berberine complexes; in particular, the PHE^560^ side chain group establishes the π–π interaction with the aromatic alkaloid. The stronger binding affinity found in the MexY-berberine complexes of PA7 and PA14 strains also involves the SER^831^ and GLN^856^ residues that correspond to GLN^831^ and PRO^856^ in the PAO1 strain.

For 13-(4-trifluoro-methyl-benzyl)-berberine (p-CF_3_), the best docking score was evaluated in the MexY–PA7 strain complex, even if it has a high affinity for both the other two MexY proteins (PAO1 and PA14). In these stabilizations, the hydrophobic residues (PHE^560^LEU^561^ALA^559^ALA^825^) and hydrophilic residues (SER^830^ in the PA7 strain, GLN^558^ and GLN^856^ in the PA14 strain, and GLN^830^ in the PAO1 strain) are involved.

In addition, 13-(2-methylbenzyl)-berberine (o-CH3) complexes are stabilized by hydrophobic residues (Table 3). In detail, the substituent aromatic ring has a hydrophobic interaction with ALA^559^, ALA^825^ in MexY–PA7 and PA14 complexes. In addition, SER^831^ is involved in a OH–π interaction and GLN^831^ in the PAO1 strain contributes to the derivative stabilization by a hydrogen bond.

Finally, 13-(4-methylbenzyl)-berberine (p-CH_3_) complexes are stabilized by hydrophobic residues and polar residues such as GLN^856^, SER^831^ in PA7, GLN^831^ in PAO1, and GLN^558^ and ARG^861^ in PA14, which are all involved in H-bond interactions.

Analyzing these interacting residues, a further observation is that mutated residues such as GLN831^(PAO1)^→SER^(^^PA7-PA14)^ THR861^(PAO1)^→ARG^(^^PA7-PA14)^ PRO856^(PAO1)^→GLN^(PA7-PA14)^ in PA7 and PA14 strain complexes stabilize the interactions with the three different derivative compounds; the most important aminoacidic substitution is related to residue PRO^825^ (PAO1), which is an alanine in PA7 and PA14 strains and a glycine in the MexY PAO1 strain. This residue is involved together with ALA^559^ in the stabilization of the substituent aromatic ring in MexY–PA7 and MexY–PA14 complexes with all three derivatives. From a previous study, it is proposed that the F610A mutation is involved in the best interaction with a doxorubicin ligand, so interactions with phenylalanine are less strong [39].

#### 3.1.3. Molecular Dynamics Results

The docked complexes underwent molecular dynamics (MD) simulations to assess the stability of the ligands’ association inside the binding site. From the analysis of MD trajectories, we can distinguish different results for each MexY–ligand complex of the three considered strains. For berberine, the stabilization is hindered by its fluctuation inside this site, which corresponds to the ligand’s partial exit as observed in the MexY–PA14 strain complex. In the PAO1–MexY complex, a transition of the ligand between the two periplasmic loops toward the inner space is observed, while in the PA7–MexY complex berberine also oscillates inside it throughout the entire simulation time. Generally, the aromatic functionalization of berberine is able to increase its stabilization within the CH1 access site, but the type of functionalizing group is important in order to maximize the interactions with the cleft’s residues.

Analyzing the RMSD curves along the MD trajectories, we can notice a different stabilization pathway (Appendix A). For the PAO1 strain, we observe a better stabilization of the o-CH_3_-berberine with respect to the p-CF_3_ and p-CH_3_ derivatives. Notwithstanding the proper orientation of the ligand within the site, the p-CF_3_ moiety is responsible for the displacement of the ligand from the fissure due to the attraction of the fluorinated group exerted by GLN^558^ and GLN^830^, which destabilize the molecular interactions with other residues inside the slit. For the p-CH_3_ derivative, even if the same initial pose of o-CH_3_-berberine was found, the methyl group’s functionalization in the para position, instead of in the ortho position, on the aromatic ring is associated with the ligand’s destabilization during the simulation, with the ring swinging and rotating around the benzylic single bond along the MD trajectory. This movement is made possible since there are no stabilizing interactions involving the p-CH3 that can fix it in a specific orientation. This different situation is ascribed to the presence of the ortho-methyl substituent, which is kept almost fixed around the same position during the whole simulation, thus leading to a higher degree of stabilization (Figure 4).

For the PA14 strain, from the MD data analysis, after the stabilization of the complex has been reached, all three derivative molecules are not located in the central position within the slit, showing a lower effect on the stabilization of complexes that is evidenced by a reduced number of specific interactions with the protein aminoacids of the binding cleft as reported from a PLIP analysis [40].

For the PA7 strain, the p-CF_3_ derivative is less stable than the o-CH_3_ one during the MD simulation due to the attraction of the fluorinated group by SER^831^, which induces the ligand to move away from the original CH1 site. Concerning the p-CH_3_ derivative, the para substitution of the methyl group leads to a slight initial destabilization due to a steric hindrance between the Ala ^559^ and Ala ^825^ residues. This steric hindrance makes the ring oscillate frequently during simulations, preventing the total stabilization of the EPI inside the cleft. A different scenario is presented by the o-CH_3_ derivative, which appears stable and strongly anchored inside the pocket; the substitution in the ortho position on the aromatic ring and the methyl group itself leads to stabilizing interactions with the derivative, which are necessary to avoid the excessive reorientation of the ligand inside the fissure, thus increasing its potential inhibitory activity.

### 3.2. Microbiological Results

#### 3.2.1. Synergistic Activity of Berberine Derivatives with Tobramycin

Preliminary MIC assays excluded any antimicrobial activity of the compounds in the tested concentration range (10–320 µg/mL) and of their solvent (1.5% DMSO) (data not shown). Then, the synergistic effect of the three berberine derivatives o-CH_3_, p-CH_3_, and p-CF_3_ in combination with tobramycin was tested against the *P. aeruginosa* strains PAO1, PA7, and PA14. The synergistic action of berberine at the active concentration of 80 µg/mL was determined as a comparison (Table 4).

Against *P. aeruginosa* PA7, showing a high degree of tobramycin resistance (MIC, 256 µg/mL), the presence of 80 µg/mL berberine induced an 8-fold decrease (from 256 to 32 µg/mL) in tobramycin’s MIC, and its derivatives o-CH_3_ at 40 µg/mL and p-CH_3_ at 320 µg/mL a 32-fold (from 256 to 8 µg/mL) and 16-fold (from 256 to 16 µg/mL) decrease, respectively. The p-CF_3_ derivative was found to be less synergic than berberine, causing only a four-fold decrease (from 256 to 64 µg/mL) in tobramycin’s MIC and was not further considered.

The two CH_3_ derivatives were then used in association with tobramycin against *P. aeruginosa* PAO1 and PA14, always causing a reduction in tobramycin’s MIC. The p-CH_3_ derivative exerted only a 2-fold decrease in tobramycin’s MIC against *P. aeruginosa* PAO1 at all tested concentrations, whereas in all other assays the compounds exerted at least a 4-fold reduction starting from the concentrations that were active against *P. aeruginosa* PA7. All the results of the checkerboard assays are summarized in Table 5.

Finally, the two most active minimal concentrations of both the o-CH_3_ (40 µg/mL) and p-CH_3_ (320 µg/mL) derivatives were tested in association with tobramycin in MIC determination assays against the Δ*mexXY* strain *P. aeruginosa* K1525 and the same strain complemented with the *mex*XY plasmid pYM004. While the mutant strain did not show any modification of its tobramycin’s MIC in the absence/presence of both compounds, the complemented strain exhibited a 2- and 4-fold decrease in the tobramycin’s MIC in the presence of the p-CH_3_ and o-CH_3_ derivatives, respectively; a behavior similar to that observed with the wild-type strain *P. aeruginosa* PAO1.

#### 3.2.2. Enhancement of Tobramycin Killing Activity by the Berberine Derivatives

The ability of the o-CH_3_ and p-CH_3_ berberine derivatives to improve the tobramycin killing activity was evaluated by killing curve assays against *P. aeruginosa* PA7 (Figure 5).

In combination with both compounds, all tested tobramycin concentrations were found to be bactericidal starting from 2 h of exposure, with a thousand-fold reduction in the inoculum. Specifically, at this time point, the combination with the o-CH_3_ derivative reduced the *P. aeruginosa*’s abundance by two logs, irrespective of the drug concentration (Figure 5A), and the combination with the p-CH_3_ derivative reduced the *P. aeruginosa*’s abundance by 4 logs compared with tobramycin alone (Figure 5B). From 4 h to the end of the experiment, the bacterial count was always ≤10 CFU/mL. Moreover, while all cultures exposed to tobramycin alone showed a CFU increase between 8 and 24 h, no CFU increase was observed for those exposed to the EPI/drug combinations.

## 4. Discussion

In this work, we evaluated the interaction between berberine and its three aromatic derivatives with the access to the periplasmic site between the periplasmic loop of TM7 and TM8 of the MexY protein, the inner transmembrane channel of the *P. aeruginosa* MexXY-OprM efflux pump, which is responsible for aminoglycoside extrusion. This site was selected because it is involved in the recruitment of lipophilic and small-sized substrates directly from the periplasmic space and it has been investigated in our previous research [11,12,41]. Indeed, a ligand that can bind strongly at this fissure could hamper the binding of natural substrates by hindering the conformational change in protomers in the MexY homotrimer in a non-competitive mechanism. The stability of the complex EPI–substrate is pivotal in order to hamper antibiotic extrusion. To evaluate the binding stability of the berberine derivatives with the different MexY variants, we performed a molecular docking and molecular dynamics investigation. The berberine derivatives showed a higher affinity than the parent compound, with more stable ligand complexes due to the presence of a mono-substituted aromatic ring, which allowed the ligands to best anchor the binding position during all of the MD simulations.

The MexY aminoacidic variants carried by the three analyzed *P. aeruginosa* strains PAO1, PA7, and PA14 were shown to influence the protein conformations and the binding stability of the simulated complexes. The docking investigation indicated that the compounds’ ability to bind more strongly depends on their conformation and orientation within the task. In particular, many interacting residues were found to be able to stabilize the berberine derivatives in a more specific way than berberine due to the presence of the aromatic substituted group. Moreover, the aminoacidic variants in these three MexY polymorphic forms can affect the binding strength, which is much higher in the case of the complexes within the *P. aeruginosa* PA7 MexY due to the presence of ALA^559^ and ALA^825^ near the aromatic ring. In *P. aeruginosa* PAO1 complexes, different residues located in the porter domain, inside the access pocket (AP), and inside the periplasmic access site affect the conformation of these regions. In fact, the differences in the aminoacidic composition mainly affect the cleft site being adjacent to the AP and representing the eligible site where EPIs bind to block the aminoglycoside extrusion. These differences in aminoacidic composition between these three MexY variants may be useful to understand the differences in the binding mode of different EPIs and their overall affinity toward the protein site. The numerous aminoacidic differences found in the sequence of PAO1–MexY compared with those found in the homologue protein carried by PA7 and PA14 could explain the quite different binding poses and the hugely different interactions exhibited by the ligands in *P. aeruginosa* PAO1, which, however, resulted in the worst score when compared with the MexY–ligand complexes in the other two strains. 

The o-CH_3_ berberine, which gave the best in vitro results, gains a different orientation inside the considered access site depending on the MexY polymorphism, as the protein aminoacidic modifications are mainly located inside and toward this location. In vitro data show an EPI activity greater than that of berberine for this derivative, which exerted a reduction in the tobramycin’s MIC up to 4 times greater than the parent alkaloid. This was particularly evident with the strain PA7, which showed the highest MIC (256 µg/mL), but was also remarkable with *P. aeruginosa* PAO1 and PA14, whose susceptibility to tobramycin was lightly or even unaffected by berberine [11]. Moreover, the compound’s active concentration was 40 µg/mL (0.070 µM), i.e., 3 times lower than that of berberine. This could be due to the stronger interaction at the binding site evidenced by the computational results. In addition, it is worth noting that within the same MexY conformation, the ligand orientation inside the binding pocket strongly depends on its functionalization (i.e., o-CH_3_ vs. p-CH_3_ vs. p-CF_3_). Accordingly, the docking and MD results clearly show that the type and the position of the functionalizing group directly influence the repositioning of the ligand and then its stability inside, as demonstrated by the experimental data.

Similar results were obtained with the p-CH_3_ derivative against *P. aeruginosa* PA7 and PA14, although only when used at a concentration greater (320 µg/mL/0.56 μM) than that of berberine (80 µg/mL/0.22 μM). Considering the molar concentration (the molecular weight of the synthesized compound exceeded that of berberine (570.36 (iodide salt) vs. 371.81 g/mol (chloride salt))), the activity of the p-CH_3_ derivative can thus be considered not so far away from that of berberine.

Time killing assays showed that the two CH_3_ derivatives were both synergic with tobramycin, with an evident bactericidal activity (a 3-log decrease in the CFU count) of the drug combination after 2 h with the p-CH_3_ derivative even when using 1/2×MIC tobramycin. Moreover, the lack of a CFU increase at 24 h when using the drug combinations suggests a role for the tested compounds in preventing the development of adaptive resistance in *P. aeruginosa* subpopulations [38]. This is pivotal for an effective bacterial clearance and the eradication of recurrent infections. These results suggest that the best ligand orientation shows the aromatic moiety oriented inside the periplasmic loops, with the lipophilic methyl group in the ortho position, thus avoiding the steric hindrance, allowing for a better adaptation of the ligand inside the lipophilic access site, and enhancing the binding affinity.

From our joint in silico and microbiological studies, it arises that the substitution of the natural alkaloid berberine with an aromatic lipophilic group leads to an evident increase in the EPI activity due to the bindings being tight and deep inside this site, with the lipophilic residues located inside. The ligands’ positioning in order to exert an inhibitory activity strongly depends not only on the nature of the functionalization but also on the polymorphism of the aminoacidic sequence of the MexY of different *P. aeruginosa* strains.

## 5. Conclusions

Polymorphic variants of MexY must be taken into account in order to rationally design new EPIs for combined antibiotic therapy and to counteract *P. aeruginosa*’s tobramycin resistance due to the drug efflux. We have shown here that polymorphisms act on ligands’ orientation due to the aminoacidic composition all around the binding site and, thus, influence the binding energy, the complex’s stability, and the dynamical evolution of binding complexes, resulting in ligand extrusion or pump blockage. The observed differences in molecules’ orientations are helpful to distinguish how a substitution in a derivative compound is better in terms of chemical features and position. In particular, for the PA7 strain, in silico models evidence differences due to polarity and the substitution sites that were also reported in vitro. In the other two strains, except for the fluorinated derivative, the other two ligands do not show substantial variations in the EPIs’ stability, which was also confirmed in vitro. The o-CH_3_ berberine was found to be the most active berberine derivative, improving the alkaloid activity and requiring lower concentrations to exert a synergistic effect; this could represent a leading compound for the design of novel and even more potent EPIs. Further in silico studies are in progress to screen a larger number of clinical *P. aeruginosa* isolates to obtain more information on MexY polymorphisms and their role in both the efflux pump substrate’s specificity and EPI effectiveness.

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
