# Peer review of "Berberine Derivatives as *Pseudomonas aeruginosa* MexXY-OprM Inhibitors: Activity and In Silico Insights"

_molecules, 2021, doi:10.3390/molecules26216644_

Round 1
Reviewer 1 Report
In this study, authors employed three different methodologies to understand the interactions of berberine and some derivatives with the Pseudomonas aeruginosa multidrug efflux system. In general the paper is well written and clear. The aim of this study and the reason to use this combined approach should be better clarified in the introduction.
The combined approach docking-molecular dynamics is well known and well described. It is not clear how authors build the force field for the berberine and its derivatives? Did they used Amber, by hand, another software? What is the protonation state of these molecules?
All other methods are well described.
In the results, authors say that the p-CH3 derivative moves and that its ring rotate. This means that in that position, there is enough room for this movement, but it is not clear if it is a continuos movement or the ring rotate in specific positions. It is not clear the "continuos rotation'. Please explain it.
Then, authors say that "showing a lower effect on the stabilization of complexes.". How did they compute this?
Also in the discussion they say: "more stable ligand complexes due". Did they computed the RMSD, RMSF of other quantities?
Author Response
In this study, authors employed three different methodologies to understand the interactions of berberine and some derivatives with the Pseudomonas aeruginosa multidrug efflux system. In general the paper is well written and clear. The aim of this study and the reason to use this combined approach should be better clarified in the introduction.
The combined approach docking-molecular dynamics is well known and well described. It is not clear how authors build the force field for the berberine and its derivatives? Did they used Amber, by hand, another software? What is the protonation state of these molecules?
Thanks very much for your comments. We missed to specify it in the methods section. Berberine and its derivatives force field parameters were built using CHARMM GUI Ligand Reader and Modeler tools which generated the parameters and topology files for the considered ligands within charmm36 ff (https://www.ncbi.nlm.nih.gov/pmc/articles/PMC5488718/). We added some sentences and a reference in the methods section. Berberine and its derivatives were in their positive charged state. (Page 4, section 2.2.3)
All other methods are well described.
In the results, authors say that the p-CH3 derivative moves and that its ring rotate. This means that in that position, there is enough room for this movement, but it is not clear if it is a continuos movement or the ring rotate in specific positions. It is not clear the "continuos rotation'. Please explain it.
Thanks, the adjective was misleading. We substituted “continuous rotation” with rotating. It means that in its positioning the aromatic moiety has not only enough space to move but this movement is also possible since there are no stabilizing interaction involving the p-CH3 which can fix it in a specific orientation; instead it is surrounded by a hydrophilic environment which induce this rotation. We added a sentence at page 9 to better clarify it.
Then, authors say that "showing a lower effect on the stabilization of complexes.". How did they compute this?
We estimated it by counting the number of interactions using PLIP (https://plip-tool.biotec.tu-dresden.de/plip-web/plip/index) which detected a much reduced number of polar and hydrophobic interactions for this MexY PA14 with respect to the other two strains. We added a sentence and a reference at page 9.
Also in the discussion they say: "more stable ligand complexes due". Did they computed the RMSD, RMSF of other quantities?
Yes, we calculated them in order to check the complex stability even if we decided not to report these data within the paper (DRMSD < 0.1 Å in the last 10 ns). In addition we checked out the difference in the stabilized pose with respect the initial one (minimized structure- equilibrated) calculating the overall DRMSD. We checked both the protein and the ligands RMSD inside the complexes and we observed a stability in orientation in the same interval, last 10 ns that for PA7 is more pronounced.
We reported ligands’ RMSD plotting as supplementary material. These results together with a lower value of free binding energy suggest a major stabilization.
We added a line at page 13 to better specify.
Reviewer 2 Report
The manuscript entitled Berberine Derivatives as Pseudomonas aeruginosa MexXYOprM Inhibitors: Activity and in silico Insights” by Giorgini et al. discusses very important aspects related to the treatment of infections caused by drug-resistant microorganisms. The studies are comprehensive including resynthesis, extended and high-level molecular modeling as well as microbiological screening. In my opinion this paper is appropriate to Molecules journal, however several points should be addressed and corrected prior the publishing, as follows:
- Authors have been searching for new EPIs useful for future therapies. Although the ability of the o-CH3 compound to increase tobramycin efficacy seem to be promising, I am not sure if berberine derivatives are enough safe to have any therapeutic future as antibiotic adjuvants. Have Authors performed any safety/druglikeness studies for the berberine derivatives?
- It is written: “Preliminary MIC assays excluded any antimicrobial activity of the compounds in the tested concentration range (10-320 μg/ml)”. How was the antimicrobial activity of these compounds excluded? How high was their intrinsic MIC?
- The following sentence is unclear and should be reworked: (Page 2) “To do this we performed an extensive comparison with berberine of their EPI activity by molecular docking and molecular dynamic simulations, followed by in vitro microbiological assays”
- Figure 3: the abbreviations “P-CF3”, “O-CF3” are incorrectly written. P and O should be lower-case letters in italic.
- Page 8: The name “4-fluoro-methyl-benzylberberine derivative” is not correct and should be improved
- Paragraph 5: “Comclusions” should be corrected
There are many minor language disadvantages along the text, in particular missing prefixes.
Author Response
the manuscript entitled Berberine Derivatives as Pseudomonas aeruginosa MexXYOprM Inhibitors: Activity and in silico Insights” by Giorgini et al. discusses very important aspects related to the treatment of infections caused by drug-resistant microorganisms. The studies are comprehensive including resynthesis, extended and high-level molecular modeling as well as microbiological screening. In my opinion this paper is appropriate to Molecules journal, however several points should be addressed and corrected prior the publishing, as follows:
- Authors have been searching for new EPIs useful for future therapies. Although the ability of the o-CH3 compound to increase tobramycin efficacy seem to be promising, I am not sure if berberine derivatives are enough safe to have any therapeutic future as antibiotic adjuvants. Have Authors performed any safety/druglikeness studies for the berberine derivatives?
Thanks for the comments. Indeed, toxicity must be taken into account. Berberine is a well tested and known natural alkaloid that is currently available as natural integrator due to its numerous healthy and multispectrum pharmacodynamic properties. Thus, many studies were undertaken to assess its toxicity (citing some: doi.org/10I.3389/fmolb.2018.00021; doi: 10.22038/IJBMS.2017.8676; doi.org/10.3390/toxins12110713). Indeed, the tested concentrations in our study lie below its toxicity, that furthermore we already verified in our previous work Laudadio et al. JNAt Prod 2019 (doi.org/10.1021/acs.jnatprod.9b00317). Besides, its synthesized derivatives were only preliminary tested predicting their ADME properties (Swiss ADME tools. Sci. Rep. 7, 42717; (2017) doi: 10.1038/srep42717) and comparing them with berberine. Considering the o-CH3 derivative, it emerges that its overall profile is very close to that of berberine, thus suggesting it could have a very similar toxicity (especially considering the pharmacokinetics and Medicinal chemistry section of the report). Anyway, at present we are planning to go further with cellular in vitro testing for this promising compound but besides we are still going further thus searching other derivatives that could improve the activity towards different MexY strains basing on the structural considerations done in this work.
2. It is written: “Preliminary MIC assays excluded any antimicrobial activity of the compounds in the tested concentration range (10-320 μg/ml)”.How high was their intrinsic MIC?
The antimicrobial activity of the three berberine derivatives was assessed by determining their MIC according to CLSI guidelines. We selected 320 μg/ml as highest concentration in order not to exceed 1,5% DMSO, to avoid possible toxic effects on eukaryotic cells. In this experimental setting, P. aeruginosa MIC towards to three compounds resulted always > 320 μg/ml, and, accordingly, bacterial growth was not inhibited when testing DMSO concentrations ≤1,5%. This was specified in the revised version of the manuscript (material and methods section 2.3.2.1. page 5).
3. The following sentence is unclear and should be reworked: (Page 2) “To do this we performed an extensive comparison with berberine of their EPI activity by molecular docking and molecular dynamic simulations, followed by in vitro microbiological assays”
Thanks very much. It was a totally unclear sentence. We reformulated it (page 2 in red)
4. Figure 3: the abbreviations “P-CF3”, “O-CF3” are incorrectly written. P and O should be lower-case letters in italic.
Done. We restyled Figure 3 and put the free binding energies in a Table (now Table 2).
5. Page 8: The name “4-fluoro-methyl-benzylberberine derivative” is not correct and should be improved
Thanks, corrected
6. Paragraph 5: “Comclusions” should be corrected
Done
There are many minor language disadvantages along the text, in particular missing prefixes.
We carefully checked the manuscript to improve the english
Reviewer 3 Report
The author use in silico design of berberine derivatives to explain the EPI potency of known berberine derivatives (Kotani antibiotics 2019). It will be interesting to use this in silico work to try to propose a more potent compound than 13-oBBer already design and tested as EPI. Exploration of the pocket interaction obtain to describ new and/or more potent inhibitor and validate it with experimental . Others substituant like o-CF3 that could be compared with 3oCH3 in term of stability, o-Methoxy to explore the size of the pocket and other position that can stabilized the berberine in the pocket. The huge work done in the modelization process and analyzis will be a plus for the work in term of novelty and design of new EPI!
Regarding the text please recheck the labeling of references: efflux systems[13] is 12, reference 34 is in the discussion and have to be 38. Recheck the editing of reference 11 and 12 one is journal of natural product the other J Nat Prod! Ref20 it's Eur J Med Chem., 26 name of the journal have to be in italic... Please recheck carefuly
some spellings errors
2.1 Synthesis
the reaction mixture was refluxed
5. Conclusions
For the readers Figure 3: is it possible to reduce the size of the full prot. and enhance the focus ligands superposition so we could see the arrow explaining the interaction or remove the ribbons and replace by the AA of interest.
Author Response
The author use in silico design of berberine derivatives to explain the EPI potency of known berberine derivatives (Kotani antibiotics 2019). It will be interesting to use this in silico work to try to propose a more potent compound than 13-oBBer already design and tested as EPI. Exploration of the pocket interaction obtain to describe new and/or more potent inhibitor and validate it with experimental . Others substituant like o-CF3 that could be compared with 3oCH3 in term of stability, o-Methoxy to explore the size of the pocket and other position that can stabilized the berberine in the pocket. The huge work done in the modelization process and analyzis will be a plus for the work in term of novelty and design of new EPI!
Thanks for your suggestions. Indeed, the main purpose of this work was to clarify the molecular basis of the EPI interaction and to assess the binding site for the three selected berberine derivatives. This computational work has never been done before and it can represent a crucial step in subsequent rational drug design. In a second phase the EPI affinity will be improved designing new potential inhibitors based on berberine scaffold and the results collected. At present, we are synthesizing other 13-aromatic substituted derivatives considering the involved aminoacidic surrounding and starting from the observed stabilization of the o-CH3 compound. In particular, we are planning to add H-donor groups in para position but leaving the o-CH3 moiety or substituting it with and -OCH3 group. Preliminary docking results were promising but they must be confirmed by molecular dynamics simulations.
Besides, we have also evaluated in silico the o-CF3 derivative binding, but as expected it did not show an high affinity, especially on PA7 MexY due to the presence of an hydrophobic specificity site around this position (o-CF3 Eb=-9.6 kcal/mol vs -10.6 kcal/mol for o-CH3). Thus, we discarded this kind of polar substitution and go further with the design.
Regarding the text please recheck the labeling of references: efflux systems [13] is 12, reference 34 is in the discussion and have to be 38. Recheck the editing of reference 11 and 12 one is journal of natural product the other J Nat Prod! Ref20 it's Eur J Med Chem., 26 name of the journal have to be in italic... Please recheck carefuly
Done . Thanks. We checked and added some new references.
some spellings errors
2.1 Synthesis
the reaction mixture was refluxed
- Conclusions
Thanks, we corrected them
For the readers Figure 3: is it possible to reduce the size of the full prot. and enhance the focus ligands superposition so we could see the arrow explaining the interaction or remove the ribbons and replace by the AA of interest.
Thanks for the suggestion. We totally restyled Figure 3 according your suggestions
Round 2
Reviewer 3 Report
All the correction done gives the article a better reading.